# The Spatio-Temporal Variability of Frost Blisters in a Perennial Frozen Lake along the Antarctic Coast as Indicator of the Groundwater Supply

**Stefano Ponti [1], Riccardo Scipinotti [2], Samuele Pierattini [3] and Mauro Guglielmin [1,*]**

[1] Department of Theoretical and Applied Sciences, Insubria University, 21100 Varese, Italy; s.ponti@uninsubria.it

[2] Technical Antarctic Unite, Energy and Sustainable Economic Development, National Agency for New Technologies, 40129 Bologna, Italy; riccardo.scipinotti@enea.it

[3] Computer Systems and ICT Development, National Agency for New Technologies, Energy and Sustainable Economic Development, 50019 Sesto Fiorentino, Italy; samuele.pierattini@enea.it

* Correspondence: mauro.guglielmin@uninsubria.it

**Abstract:** Remote sensing, and unmanned aerial vehicles (UAVs) in particular, can be a valid tool for assessing the dynamics of cryotic features as frost blisters and to monitor the surface changes and the sublimation rates on perennially frozen lakes that host important ecosystems. In this paper, through the use of these remote sensing techniques, we aim to understand the type of groundwater supply of an Antarctic perennial frozen lake that encompasses two frost blisters (M1 and M2) through the temporal analysis of the features' elevation changes (frost blisters and lake ice level). The frozen lake is located at Boulder Clay (northern Victoria Land, Antarctica). We relied on several photogrammetric models, past satellite images and ground pictures to conduct differencing of digital elevation models, areal variations and pixel counting. In addition, in situ measurements of the ice sublimation or snow accumulation were carried out. The two frost blisters showed different elevation trends with M1 higher in the past (1996–2004) than recently (2014–2019), while M2 showed an opposite trend, similarly to the ice level. Indeed, the linear regression between M2 elevation changes and the ice level variation was statistically significant, as well as with the annual thawing degree days, while M1 did not show significant results. From these results we can infer that the groundwater supply of M1 can be related to a sublake open talik (hydraulic system) as confirmed also by pressurized brines found below M1, during a drilling in summer 2019. For M2 the groundwater flow is still not completely clear although the hydrostatic system seems the easiest explanation as well as for the uplift of the lake ice.

**Keywords:** frost mounds; frozen lakes; structure from motion; Antarctic coasts

## 1. Introduction

In coastal areas of extremes environments like Arctic or Antarctica, remote sensing techniques are frequently used due to the terrain inaccessibility [1,2]. In particular, structure from motion (SfM) photogrammetry has been largely used especially for improving beach 3D reconstructions [3,4], although on Antarctic coastal areas SfM is usually used for monitoring sea ice, e.g., [5], ecosystems, e.g., [6] or moss beds, e.g., [7], but rarely for topographic changes, e.g., [8] or periglacial landforms mapping, e.g., [9]. Nevertheless, in the ice-free coastal areas of continental Antarctica, periglacial and cryotic features such as thermal-contraction-crack polygons, ice-wedges, perennially frozen lakes including frost mounds [10,11] and rock glaciers [12] are widespread.

Seasonal frost mounds are small permafrost features that have three different types (frost blister, icing blister and icing mounds) according to their structure [13,14]. These landforms are common in the Arctic, e.g., [15–18] but poorly studied in Antarctica [10,19,20], indeed only one paper regards their spatio-temporal variations although qualitatively [10,11,19,20].

The structure of frost blisters is not univocal, in fact the intrusive ice may form through a closed or open system of groundwater freezing [14,21] that could exert enough pressure on the overlying sediments when it freezes. In the first case, water is confined and the hydrostatic pressure is an effect of free water that does not encounter the freezing front. In the case of open-system, groundwater is able to flow and to generate a hydraulic potential and large pressures [21]. As a consequence, the understanding of frost blisters' structure and genesis should be related to the assessment of the groundwater circulation.

Permafrost hydrology and groundwater systems recently increase the attention of scientific community also in Antarctica [22–25], but rarely concerning frost mounds [10,20]. Indeed, groundwater flow has been studied as a consequence of the surficial hydric balance that causes the infiltration of meltwater [23] also through snow redistribution [24]. However, the hydrology of Antarctic lake systems also depends on the formation and sublimation of ground ice in the catchment [22], the presence of brines [22,25] and the sublimation of lake ice [26]. It is a matter of fact that the hydrology of perennially frozen Antarctic lakes can be linked to lake ice icing blisters [10] or to pressurized brines beneath pingo-like features [20]. However, growing rates [27] or vertical movements of frost mounds [28] have been measured in Canadian Arctic, while, according to our knowledge, no scientific papers on this topic related to Antarctica have been published. Therefore, the monitoring of the temporal variations of the frost blisters could be also useful to understand the mechanism of water supply (closed vs open system, [29]).

Due to the connection of Antarctic frost mounds with frozen lakes, the hydrology of the latter should not be excluded for the assessment of the whole system hydric balance. Indeed, the talik type and its water supply are key factors in the analysis of permafrost impacted systems [30] and frost mounds [10,20].

The hydric balance and the recharge system in the cryotic environment could only be understood with a quantification of the role of sublimation and snow accumulation that are generally more important than the melting [26,31–34]. Moreover, the climatic importance of permafrost-related groundwater systems has been demonstrated in Antarctic lakes [10,33–36] and this will help in figuring out the future scenarios [37,38].

In this paper we focus on one of the perennially frozen lakes studied by Guglielmin et al. [10] and recently investigated also regarding its microbiological ecosystems [39,40]. In particular, we quantify the temporal variation of the two frost blisters occurring in this lake and the spatio-temporal variations (i.e., sublimation and accretion) of the ice surface in order to assess the origin of the water supply that is responsible of the lake-system dynamic.

## 2. Materials and Methods

### 2.1. Study Area and Climate Analyses

The study site is a perennially frozen lake (74.7466°S, 164.0197°E, 205 m a.s.l.) located in an area called Boulder Clay, close to Mario Zucchelli Station (MZS) in northern Victoria Land (Antarctica) (Figure 1). The lake is a dry-based type and, in summer, melting can occur at its bottom, thus generating saline water [10]. On its surface, two icing blisters are visible (black triangles Figure 1), while two frost blisters (M1 and M2), developed from the basin, cross the ice surface [10] (Figure 1).

This ice-free area is 6 km south of MZS and it has a southeastern aspect with a general slope smaller than 5°. The climate is characterized by a mean annual air temperature (MAAT) of −14.1 °C if considering the climatic series of this study (1990–2019) and a scarce solid precipitation of 100–200 mm water equivalent (w.e.). [41]. Despite the low precipitation, snow drift is very important because the strong winds may favor a snow cover with a high variability [42].

The area consists of glacial ablation-sublimation till overlaid on a body of a dead glacier [43]. The till matrix is generally silty sand with confined zones of clayey silt [42]. The vegetation cover in this area is scarce and composed by mosses and epilithic lichens [42,44].

Permafrost in this area is continuous and 420–900 m thick [45]. Active layer ranges between 23 and 92 cm with a thickening trend of +0.3 cm per year [42]. Here, numerous surface features were encountered, such as perennially ice-covered ponds, frost-fissure polygons and frost mounds (frost blisters, icing blisters) [10,11,43,46].

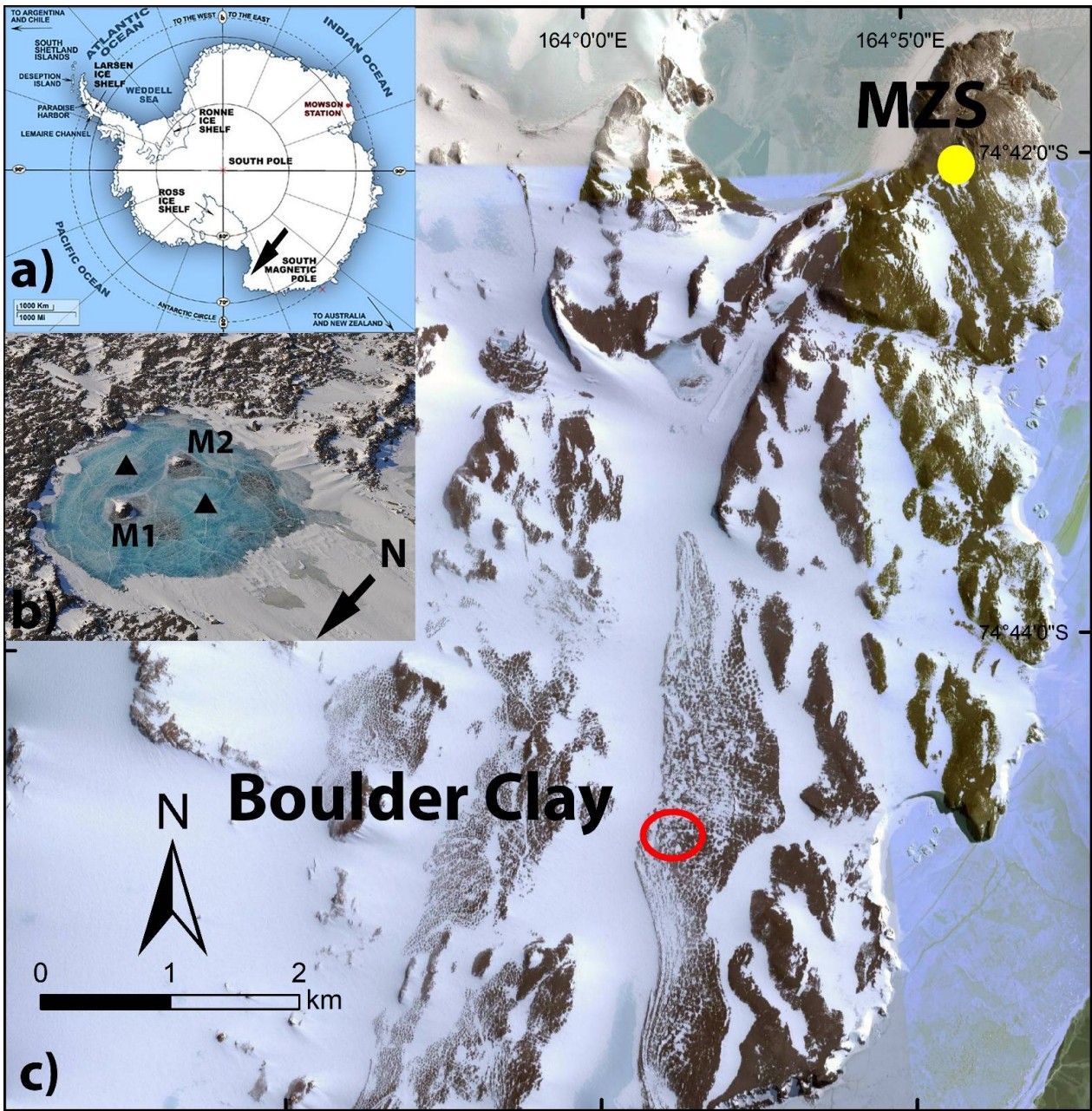

**Figure 1.** Location of the study area. (**a**) Location of Northern Victoria Land (black arrow), (**b**) helicopter view of the two frost blisters and the two icing blisters (black triangles), (**c**) Boulder Clay satellite imagery (Google Earth® satellite imagery of 2014). The lake is approximatively 6 km south of the Italian station (MZS) and marked by the red circle. The yellow dot indicates the automatic weather station (AWS) "Eneide".

The calculation of mean air temperatures for different spans of time (daily, monthly, seasonally, yearly) and the thawing degree days (TDD) following Molau and Mølgaard [47] allowed to obtain: (a) mean annual air temperatures (MAAT), (b) mean summer (December, January, February, DJF) air temperatures (MSAT), (c) mean winter (June, July, August, JJA) air temperatures (MWAT) and annual thawing degree days (ATDD) [48]. Hourly air temperatures were collected from the closest automatic weather station (AWS) called

"Eneide" (74.70°S 164.10°E; 92 m a.s.l., Figure 1) from 1990 to 2019 (Data and information were obtained from 'MeteoClimatological Observatory at MZS and Victoria Land' of PNRA–http://www.climantartide.it). Moreover, to assess the climatic component that affect the surface changes of the two frost blisters, we compared the elevation with the ATDD of the previous year through a linear regression analysis. In addition to estimate the total sublimation, a linear regression between the ice level variation collected in situ during December 2018 and a climatic index (ATWS) was computed and extrapolated to the entire year. ATWS is the multiplication of the mean air temperature for the mean wind speed considering the period including the three days before the ice level measurement.

*2.2. Photogrammetric Analysis*

Four photogrammetric surveys were conducted during the summers of 2017–2018 and 2018–2019 (A refers to 08/11/2017, B to 03/12/2018, C to 30/12/2018 and D to 21/11/2019) to create the correspondent digital elevation models (DEMs) and the related orthoimages of the lake with Agisoft Metashape 1.5 (Table 1). Due to the harsh weather conditions and logistical constraints, it was not possible to guarantee the same date of survey along the study period. However, the surveys represent the warmest months of the year. In addition, during 2018 two surveys were conducted to assess intraseasonal changes. For (A), a helicopter survey was conducted to take convergent images from a circular orbit [49]. For (B), (C) and (D), the survey consisted of a drone flight with a nadir acquisition [50]. The surveys guaranteed a general image overlap of >50%. The drone was a hexacopter aircraft, a customized version of the SR-SF6 model (Skyrobotics, Italy). Ground markers were selected and georeferenced with a Leica GPS station comprising a GS10 receiver, a CS10 controller, an AS10 antenna and a real-time-kinematic (RTK) modem (horizontal and vertical accuracy of 8 and 15 mm, respectively). Some of the markers were used as ground control points (GCPs) and the remaining as check points [51]. The choice of GCPs was constrained by the quality of the images and the characteristics of each photogrammetric survey are summarized in Table 1.

**Table 1.** Characteristics of the cameras (resolution), the markers (number) and the accuracy of the four 3D models obtained from the photogrammetric reconstructions.

| Survey | Date | Camera | Markers | GCPs | Check Points | Resolution (cm) | Reprojection Error of Tie Points (px) | Reprojection Error of Markers (px) | Vertical Error of Check Points (cm) |
|---|---|---|---|---|---|---|---|---|---|
| A | 08/11/2017 | Honor 9 Smartphone—20 MP | 9 | 5 | 1 | 8.1 | 1.0–3.6 | 1.8–3.2 | 2.3 |
| B | 03/12/2018 | Sony QX1—20 MP | 21 | 13 | 7 | 0.7 | 0.3–1.0 | 0.3–0.6 | 2.0 |
| C | 30/12/2018 | Sony QX1—20 MP | 20 | 13 | 3 | 0.6 | 0.3–1.5 | 0.2–1.0 | 1.0 |
| D | 21/11/2019 | Sony QX1—20 MP | 15 | 12 | 3 | 0.7 | 0.5–0.9 | 0.3–0.9 | 1.5 |

The bundle adjustment that yielded the 3D models from multiview stereo images provided a good resolution of DEMs ranging between 0.6 and 8.1 cm depending on the camera and the flight mission settings.

The calculations conducted on the DEMs obtained from the above-described models always respected the same type of survey: (A) compared to (B), (B) compared to (C) and (C) compared to (D). This provided the most recent assessment of elevation changes of the lake surface and frost blisters (Table 2).

In order to get a longer period for temporal changes, several views of the (B) dense cloud in Metashape were captured in order to match the same perspective and distance of ground-level images taken in previous Antarctic campaigns (1996, 2002, 2004, 2014). (B) was selected as reference dense cloud because it was the most snow-free and less biased model of 2018, as well as the year with the highest frequency of analyses. This provided couples of images ready to assess differences between 2018 and each previous campaign (Figure 2).

**Table 2.** Multidimensional differences of the frost blisters dimensions and the ice level during the study period. The pixel analysis shows the reconstructed elevations for the period 1996–2014 graphically shown in Figure 4. The DEMs shows the real elevations for the period 2017–2019. Area and volume show the dimensions of M1 and M2 calculated from the aerial images, orthophotos and DEMs in different years. "Ice" from 2017 to 2019 refers to the mean change of the internal ice area near the frost blisters.

| | Mean Elevation from Pixel Analysis (m a.s.l.) | | | Mean Elevation from DEM (m a.s.l.) | | | Area (m²) | | Volume (m³) | |
|---|---|---|---|---|---|---|---|---|---|---|
| | **M1** | **M2** | **Ice** | **M1** | **M2** | **Ice** | **M1** | **M2** | **M1** | **M2** |
| 1996 | 206.067 | | 205.468 | | | | | | | |
| 2002 | 206.231 | 205.489 | 205.036 | | | | | | | |
| 2004 | 206.198 | 205.229 | 205.196 | | | | 32.0 | 21.3 | | |
| 2009 | | | | | | | 19.6 | 0 | | |
| 2011 | | | | | | | 15.9 | 1 | | |
| 2014 | 205.569 | 205.058 | 204.878 | | | | 18.2 | 16.3 | | |
| 2017 (A) | | | | 205.913 | 205.762 | 205.563 | | | 1.93 | 0.95 |
| 2018 (B) | | | | 205.997 | 205.801 | 205.435 | 20.7 | 23.6 | 2.35 | 1.24 |
| 2018 (C) | | | | 206.058 | 205.874 | 205.468 | | | 2.76 | 1.49 |
| 2019 (D) | | | | 205.98 | 205.807 | 205.29 | 22.3 | 21.5 | 2.35 | 1.17 |

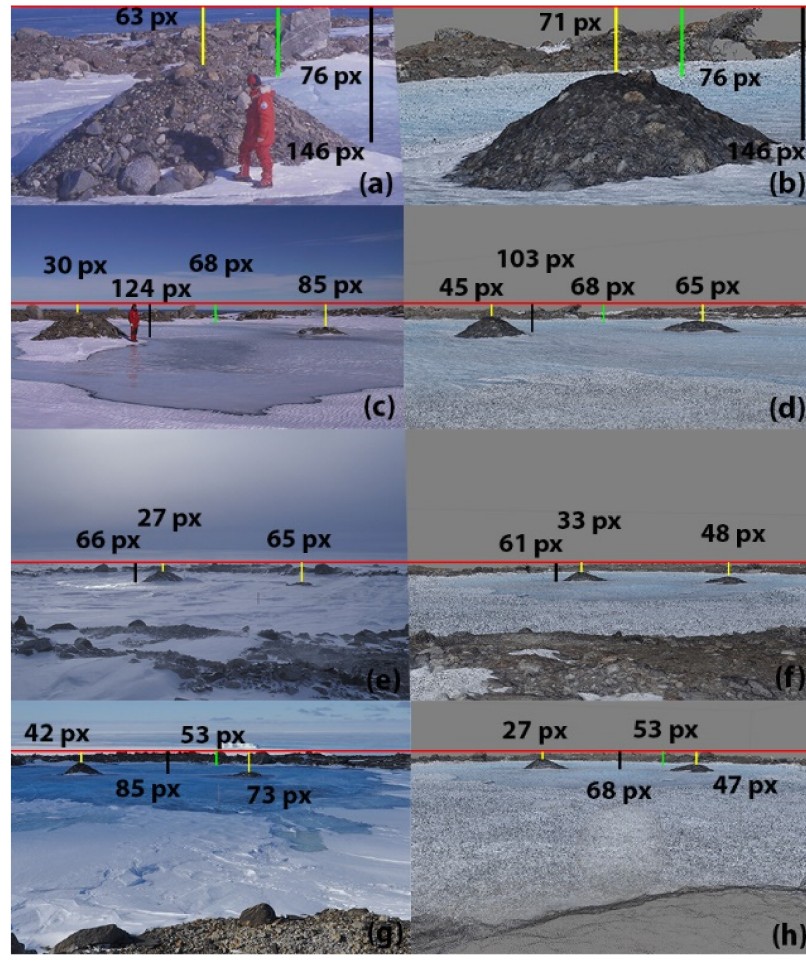

**Figure 2.** Comparison of heights in pixel between the frost blisters M1 (left), M2 (right) and ice references in different years. The matching accounts for images taken in 1996 (**a**), 2002 (**c**), 2004 (**e**), 2014 (**g**) and each respective shooting position of the (**c**) on the 30/12/2018 (**b,d,f,h**). The red line links the external fixed points (boulders), the yellow bars match the frost blisters heights, the green bars the lake ice level at its eastern border, the black bars the internal ice level.

The measurement of the frost blisters' heights in previous campaigns was conducted by pixel counting in Adobe Photoshop. A background layer of 300 dpi was used to compare the two images in order to have the same amount of pixels per object in both the pictures. A horizontal line was drawn to align the top of stable boulders outside the lake borders and subsequently the pixel distance of the two frost blisters' tops from the horizontal line was measured. This yielded an assessment of height difference relative to the outer part of the lake along a temporal scale. This method partially followed the one used in the time-lapse photography of van Everdingen [27].

The ice level at the base of each frost blister has been variable due to sublimation or, rarely, melting (only in January 2002). This situation was useful to additionally calculate the variation of the ice surface from the horizontal line (i.e., the icy borders of the lake and internal areas like the icing blister). Finally, by converting the number of pixels in centimeters with the known height of selected objects, it has been possible to (a) quantify (in cm) the variation of both the ice surface and frost blisters' heights during the years, and (b) reconstruct their original elevation (in m a.s.l.) in each year by starting from the most recent surveys' known elevations (DEMs) (Table 2).

Beside the elevation change of the frost blisters, also an areal quantification was conducted. The available chronological sequence (2004, 2009, 2011, 2014) of the aerial images of the lake was downloaded from Google Earth Pro® and georeferenced in ArcGIS 10.3. Here, an areal comparison of the frost blisters in 2004, 2009, 2011 and 2014 plus the most snowfree orthophotos (A), (B) and (D) was possible.

In addition, a volumetric temporal reconstruction of the frost blisters was provided. The DEMs obtained from (A), (B), (C) and (D) were cut off at a fixed elevation to maintain the top part of the frost blisters (that is the area not affected by the snow drift/accumulation) and the volume calculated for each year. This was conducted by summing the volume of each parallelepiped having a resampled base (pixel) dimension of 8.1 × 8.1 cm (Table 2).

### 2.3. DEM Analysis and Sublimation Rates

For a topographic analysis, DEMs and orthoimages obtained from (A), (B), (C) and (D) were treated in ArcGIS 10.3. All the DEMs yielded a quantification of the blisters and ice change both annually (A, B, D) and seasonally (B, C). Since the snow cover variability is mainly related to snow drift, only the ice surface was considered to undergo sublimation. Indeed, ice sublimation here is largely the main process that could reduce the ice surface level because meltwater was observed only once during January 2002, except for a slight melting along the shorelines for a few days in a couple of other years in the last 20 years. Therefore, only the snow-free area of (A), (B), (C) and (D) was considered for the calculation.

Indeed, the most accurate DEMs (B, C, D) (lowest error values, Table 1) were useful to assess the sublimation rates of the ice surface at midseason 2018 (B–C) and during one year 2018–2019 (C–D) through differencing of DEMs, while orthophotos were used for the characterization of the surface types (snow, ice, frost blisters).

The calculation of error propagations in the DEM operations followed the method described by Milan et al. [52] and Brasington et al. [53] at 95% of confidence interval. It is showed in Equation (1):

$$E = 1.96 \sqrt{((\sigma_1)^2 + (\sigma_2)^2)} \tag{1}$$

where E is the final error, while $\sigma_1$ and $\sigma_2$ are the standard deviations of the elevation error of the two DEMs, respectively.

In addition, field measurements of surficial snow deflation/sublimation or ice sublimation on the lake surface were carried out through the installation of 40 white plastic tubes (40 cm long, 1.5 cm of diameter), inserted vertically at 15 cm below the surface and distributed on the lake statistically representing the different surface types. Four measures of the above-surface length of each tube were recorded and averaged [10] with a ruler every 10 days from 11/11/2018 to 16/01/2019. In this way, the seasonal surficial gain (for snow) or loss (for ice and snow) was calculated by summing the vertical differences at each tube.

## 3. Results

### 3.1. Climate

In the analyzed period (1990–2019) the MAAT was −14.1 °C, while MSAT and MWAT were −2.7 and −21.0 °C, respectively. Similarly, the mean ATDD was very low (26.8 °C). All the trends were slightly positive and statistically significant ($p < 0.05$), except for MWAT that showed a negative trend not statistically significant ($p > 0.05$) (Figure 3). In detail, the trends are shown by the linear regression between the climate parameters and the time as follows (year = x): MSAT = 0.0393x − 3.3268 ($R^2$ = 0.18), MAAT = 0.0339x − 14.591 ($R^2$ = 0.2), MWAT = −0.0046x − 20.943 ($R^2$ = 0.001), ATDD = 0.6061x + 17.393 ($R^2$ = 0.16). The linear regressions that explain the relations between the blisters' elevations with the ATDD and the ice sublimation with the ATWS have been treated in the discussion.

### 3.2. Frost Blisters

The comparison of all frost blisters images available is reported in Figure 2 where on the left column there are the different frost blisters images available, while on the right column there are the different views of the dense cloud (B) (3/12/2018) that match the shooting position of the left column (Figure 2).

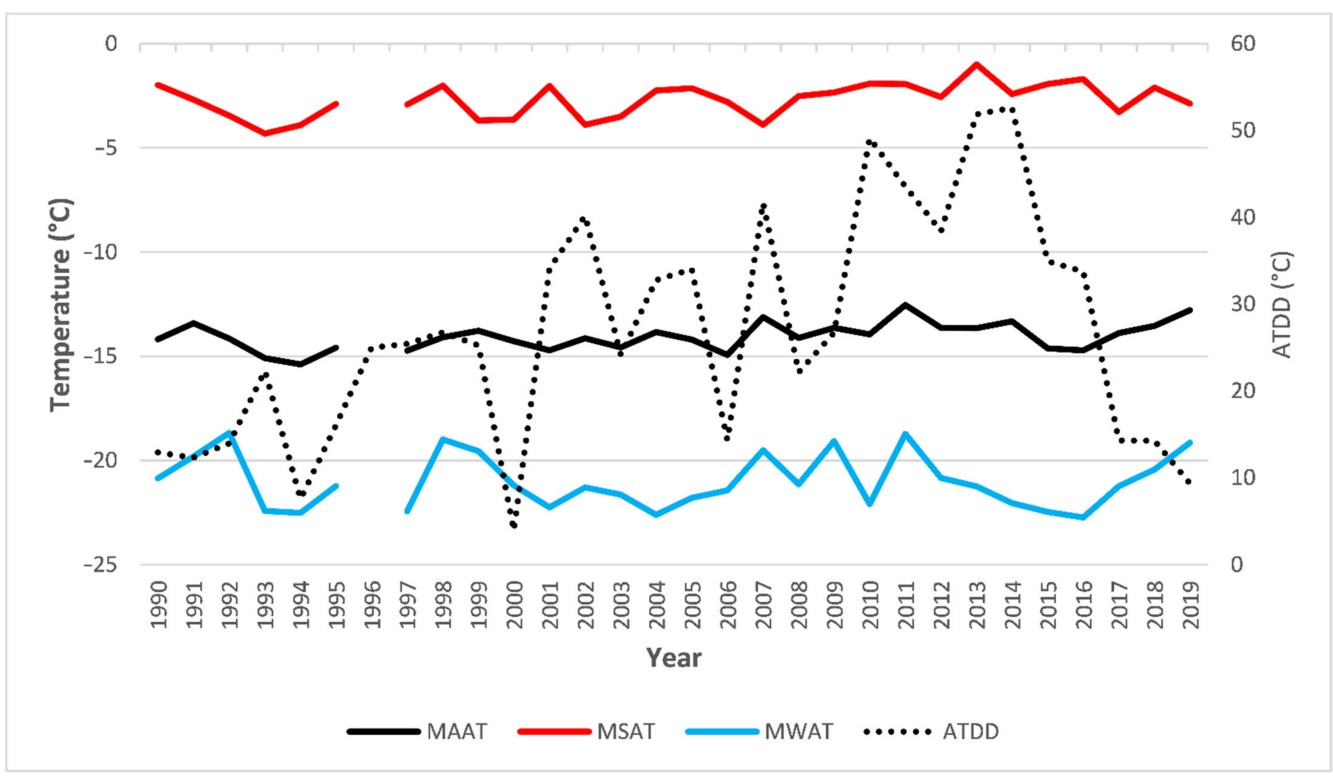

**Figure 3.** Trends of the principal climatic parameters for the period 1990–2019 at the closer available automatic weather station (AWS) "Eneide" 5 km far from the analyzed lake at Boulder Clay. Legend: MAAT = mean annual air temperature; MSAT = mean summer air temperature (DJF); MWAT = mean winter air temperature (JJA); ATDD = annual thawing degree days.

The top of the two frost blisters varied independently: Figure 2a M1 and M2 were both higher in 2018 respect all the previously available years except for 2014 (Figure 2e,f). It is noticeable that M1 have been bigger in the past (1996–2014) even though it constantly deflated in that period and inflated during 2014–2018. Likewise, M2 constantly decreased during 2002–2014 to then increase during 2014–2018. It is remarkable how the difference between M1 and M2 reached the maximum in 2004 with ca. 97 cm and afterwards decreased to ca. 17 cm in 2018 (Table 2). Although M1 and M2 variations followed a similar trend, they displayed no statistically significant relations.

In addition, the lake ice level at the eastern border of the lake (green bar) was constant through the comparisons (except for 2004 when the lake ice was hidden by the snow cover), while its level in correspondence with ice cracks (black bar) near the frost blisters (internal ice level) was higher in 2018 than in all the other reference years (except for 1996) and more pronounced in the 2002–2018 comparison (Figure 2c,d). If we consider the calculated elevations (Table 2), 2014 was the year with the lowest height both for M1 and M2 and for the internal ice. Indeed, conversely to the blisters, the internal ice surface pulsated (rose and subsided) every year since 1996 until 2019. It is also interesting to notice that between 2014 and 2017 the three features underwent a bigger change compared to previous and following years (+34.4, +70.4 and +68.5 cm for M1, M2 and Ice, respectively) but probably this can be related to the change of method (DEM differencing) (Table 2, Figure 4).

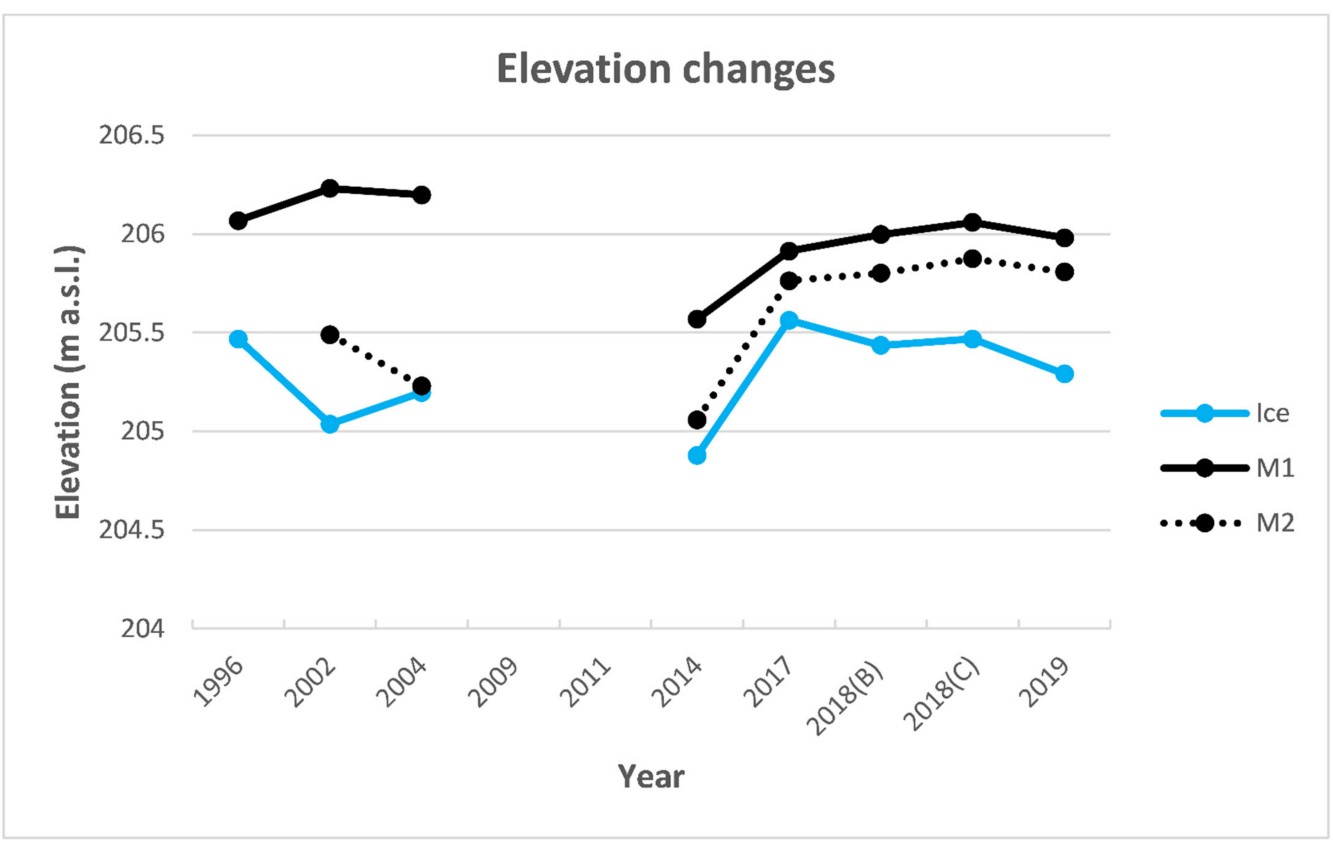

**Figure 4.** Elevation changes (m a.s.l.) between 1996 and 2019 (see also Table 2). From 1996 to 2014 the pattern relied on pixel analysis, while from 2017 to 2019 on DEM differencing. This could explain the large increase between 2014 and 2017.

The areal variation of M1 and M2 was assessed by the aerial images of Google Earth® and the orthophotos. The temporal variability of M1 was smaller than M2, indeed, M1 ranged between 15.9 m² in 2011 and 32.0 m² in 2004 while M2 ranged between 0 m² in 2009 and 23.6 m² in 2018 (Table 2).

The areal extend of the blisters showed a pulsating change that followed the elevation change. One discrepancy with the elevation change (M1 between 2018 and 2019) could be explained by the variability of snow accumulation on the mound that biased the analysis, even though only the most snow-free images and orthophotos were considered. However, the areal analysis showed changes during 2009 and 2011 that were without vertical information: interestingly, M2 disappeared during those years (0–1 m²) and reached the maximum in (B) (23.6 m²), while M1 in 2004 (32 m²). The interesting disappearance of M2 during 2009–2011 has been a result of a combination between a blister extent reduction and an ice level rise in the same span of time (2004–2009).

Similarly, the volumetric analysis agreed with the elevation changes and also with the areal analysis except for M1 (2018–2019), where a volumetric consistency (2.35 m$^3$) matched an areal increase (20.7 to 22.3 m$^2$). Anyway, the maximum volumes were reached in (C) with 2.76 and 1.49 m$^3$ for M1 and M2, respectively (Table 2).

### 3.3. Sublimation

Through DEMs differencing (B–C) and (C–D), the absolute variation of the lake surfaces was calculated. As a result, during December 2018, frequent snow drifts occurred and averagely produced an accumulation of 14.9 cm of snow. Accretion also occurred at the surfaces that changed between ice and snow depending on the snow drifts (i.e., ice/snow) (8.6 cm) and even on ice (3.6 cm). After one year, both the average snowpack and the internal ice level were similarly sublimated and/or blown away in the case of the snow (−12.3 and −18.1 cm, respectively) (Table 3).

**Table 3.** Difference of digital elevation models (DEMs) that show the extents (%) and statistics (means and standard deviations) of accretions (positive values) and sublimation (negative values) of the different surface types during the month of December 2018 (B–C) and between 2018 and 2019 (C–D). The accuracy indicates the propagation of errors during the differencing.

| | Area (%) | | Mean Change (cm) | | St. Dev. (cm) | | Accuracy (cm) | |
|---|---|---|---|---|---|---|---|---|
| | 2018 | 2019 | 2018 | 2019 | 2018 | 2019 | 2018 | 2019 |
| Ice | 21.9 | 48.7 | 3.6 | −18.1 | 2.5 | 6.1 | | |
| Snow | 51.3 | 51.3 | 14.9 | −12.3 | 8.9 | 9.2 | 4.4 | 3.5 |
| Ice/Snow | 26.8 | - | 8.6 | - | 5.2 | - | | |

Considering the whole summer season 2018–2019 among the 40 tubes, on average 19 represented the ice surface, eight the snow and 13 the ice/snow changing surfaces, while, considering only December 2018, the classification was slightly different: 13, 10 and 17, respectively (Table 4). The highest sublimation was recorded between 11 and 22 November 2018, when the mean loss accounted for −1.7 cm of ice, −2.2 cm of snow and −2.0 cm of ice/snow (not shown). Similarly, at the end of the season, the mean of the cumulative variations at each tube showed close values. In fact, the total losses were maximum for ice/snow (−2.4 cm), followed by snow (−2.0 cm) and ice surfaces (−0.7 cm) (Table 4). It is also interesting to notice that a) the largest accretion/sublimation (big dots) occurred on the snow and more distant to the frost blisters and b) the distribution of the lowest accretion/sublimation (little dots) was randomly distributed (Figure 5).

Locally, the vertical variations recorded in situ (tubes = y) and remotely-sensed (DEMs = x) matched quite well during December 2018 (B–C) for the most Table 4 tubes, according to the linear regression y = 69.9x − 3.12 (R$^2$ = 0.72, $p$ < 0.001). Conversely, no relations were found between the seasonal in situ vertical variations and the DEMs difference.

**Table 4.** In situ losses (negative values) and gains (positive values) of the different surface types during the summer season 2018–2019.

| | Min (cm) | Max (cm) | Mean (cm) | St. Dev. (cm) |
|---|---|---|---|---|
| Ice | −4.3 | 3.4 | −0.7 | 1.7 |
| Snow | −12.0 | 10.8 | −2.0 | 8.6 |
| Ice/Snow | −13.1 | 7.9 | −2.4 | 5.3 |

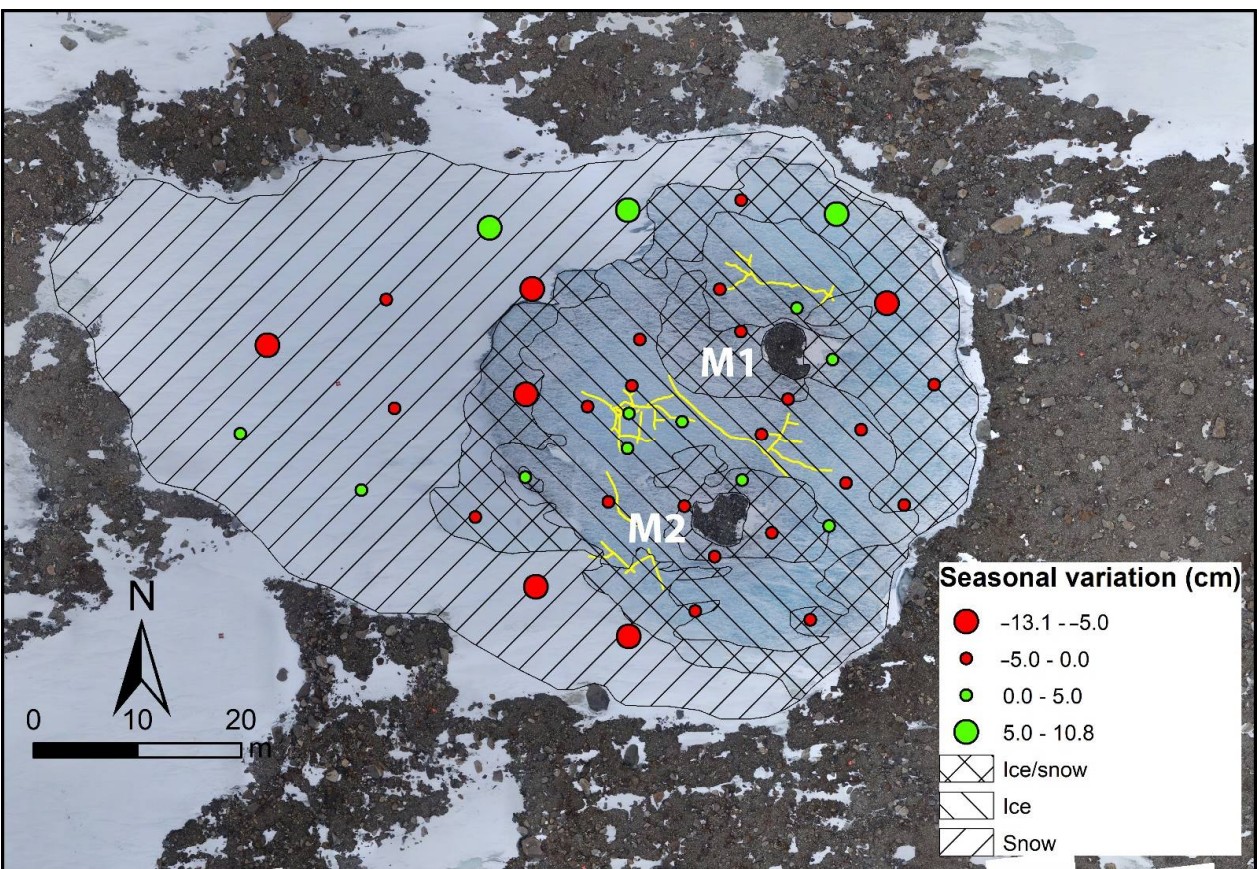

**Figure 5.** Seasonal snow and ice dynamic at the study site. The orthophoto has been obtained with the survey on 3/12/2018 while the surface type classes remain constant during all the summer 2018–2019. In addition, punctual cumulative height variations recorded at each tube from 11/11/2018 to 16/01/2019 are reported. Colors refer to sublimation (red) and accretion/accumulation (green), while dots diameter refers to high values (big) or low values (small). Yellow lines indicate the major dilation cracks on the ice surface.

## 4. Discussion

### 4.1. Frost Blisters and Lake Ice Surface Variations

Despite the different size of M1 and M2, their changes followed a similar trend and generally consistent with elevation and volumetric changes while areal changes were sometimes decoupled with the formers and this probably reflects the higher accuracy of the first two monitoring methods. Indeed, areal changes may have not only been correlated with the blisters' accretion or deflation, but also with the snow accumulation around them and the ice level dynamic.

The similar trend of elevation changes (Figure 4) showed that M1 was always the highest, although the elevation differences between M1 and M2 are largely variable, ranging between ca. 1 m in 2004 and 0.17 m in 2019. M2 results were more variable over time than M1 (Figure 4 and Table 2). Moreover, it is remarkable that during December 2018 the frost blisters showed a differential accretion of 2.0 and 2.4 mm day$^{-1}$ for M1 and M2, respectively, both greater than the lake ice surface accretion (1.2 mm day$^{-1}$). It is not surprising that these values are lower than the values recorded in the Arctic where the accretions could also reach 0.55 m day$^{-1}$ [27] because of the shallower active layer and the shorter and much colder summer in continental Antarctica. Furthermore, dilation cracks on the ice surface (Figure 5) and the increase of the lake ice level suggest that there also an uplift of the lake surface [10].

During the whole examined period, the pattern of ice level changes has been even more variable than that of the blisters. This could be explained by the debris cover that can

reduce the ice sublimation within the permafrost of the two frost blisters compared to the sublimation on the lake ice or on the snow as known in literature [54]. For example the sublimation measured on the till in the Dry Valleys was around 0.22 mm year$^{-1}$ [54], while on glacier ice of continental Antarctica was up to 40 cm year$^{-1}$ [55] and on perennially frozen lake the annual loss of ice ranging between 0.64 and 0.99 m year$^{-1}$, e.g., Dugan et al. [26].

Only the M2 elevation changes displayed a good linear regression with ice level changes during the study period. Indeed, in Figure 6 it is possible to see that only M2 responded to the ATDD displaying a statistically significant negative trend with its height changes ($R^2 = 0.85$, $p < 0.01$) and with the ice level changes following a statistically significant positive trend ($R^2 = 0.71$, $p < 0.05$). Our data confirmed the correlation between summer TDD with the ice loss already demonstrated on perennially frozen lakes of the Dry Valleys [35,56]. Moreover, the differences between the two blisters suggest that M2 and M1 have probably two different water sources because M1 seems neither related to the ice level neither to the ATDD. These facts suggest that both M2 and the ice level respond to the same climatic index (ATDD) that can be considered a proxy of the water availability because indicate the amount of energy available for the snow melting.

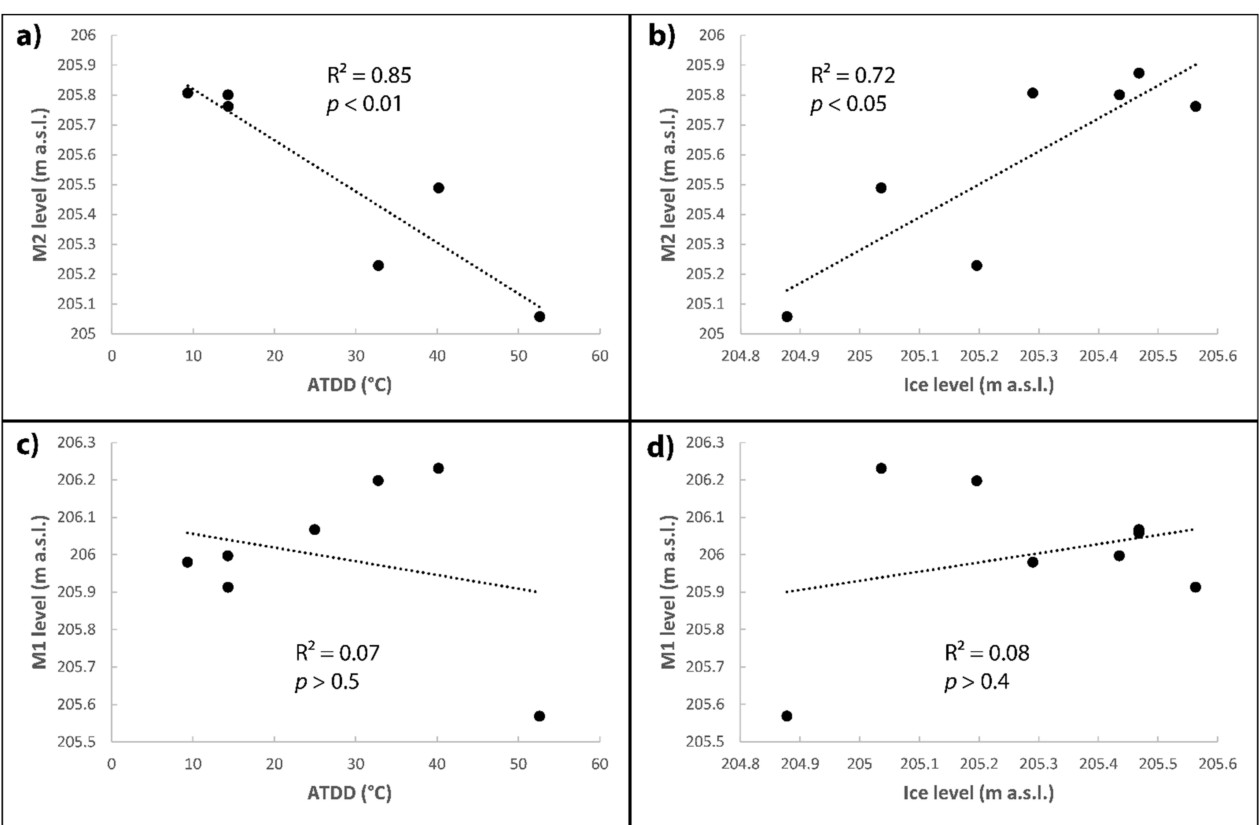

**Figure 6.** Linear regression between the frost blisters' elevation change and the ATDD (**a,c**) and the ice level (**b,d**). M2 elevation changes are plotted against the ATDD of the previous year (**a**) and then against the ice level elevation (**b**). Similarly, M1 elevation changes are plotted against the ATDD of the previous year (**c**) and then against the ice level elevation (**d**).

### 4.2. Lake Ice and Snow Surface Sublimation

The water balance of this study consisted in the monitoring of the ablation of snow and ice and the accumulation of wind-blown snow. Since surface meltwater has never been observed except in 2002 (M.G. personal communication), we assume that the snow and ice ablation mainly occurred via sublimation. Due to the high variability of snow accumulation, but especially removal (i.e., snowdrift sublimation, [57]) because of the katabatic wind action at the study area [32], the sublimation assessment was possible

only at the ice surface. However, ice sublimation plays a more important role than snow sublimation due to its higher density and so higher water equivalent, but also its lower albedo and temperature [58]. From the in situ measurements, the average loss of 0.7 cm resulted to be rather small, especially considering that the summer sublimation of east Antarctic coasts contributes more largely to the ice ablation compared to winter [54,56]. It is likely that since our summer period (NDJ) did not follow the common Antarctic summer (DJF) [33], the February's sublimation was lost. However, it is also true that the plastic tubes had no external reference points and represented a relative variation, thus inducing in underestimations or overestimations due to some possible tube sinking (ice melting) or heaving (meltwater refreezing), respectively. Conversely, by looking at the absolute remotely-sensed data, from late December 2018 (C) to November 2019 (D) a greater ice ablation of $-18.1$ cm was observed, in line with what found by Doran et al. [59] in the Dry Valleys (15–50 cm yr$^{-1}$) or Dugan et al. [26] (7–21 cm yr$^{-1}$), but slightly lower than what found by Guglielmin et al. [10].

It is well documented that ice sublimation increases when wind speed and air temperature increases, e.g., [26,34,58], but, as demonstrated by Andersen et al., the wind is more effective than air temperature [33] at least in Antarctica. Here, we obtained a statistically significant linear regression between the ice sublimation and the ATWS (Figure 7). Through this regression we calculated the annual sublimation that was very similar (18 cm) to the remotely-sensed measured one (18.1 cm). Hence, at Boulder Clay, winter ice sublimation has a greater impact than summer sublimation (because wind speed is higher on average and air temperature strongly lower) and this is in accordance to Andersen et al. [33]. Moreover, the maintenance of a high winter sublimation could be related to a scarce snow cover of the lake that would protect the ice surface [34,54,60,61], even though the subsurface sublimation of ice/snow is possible, e.g., [61,62].

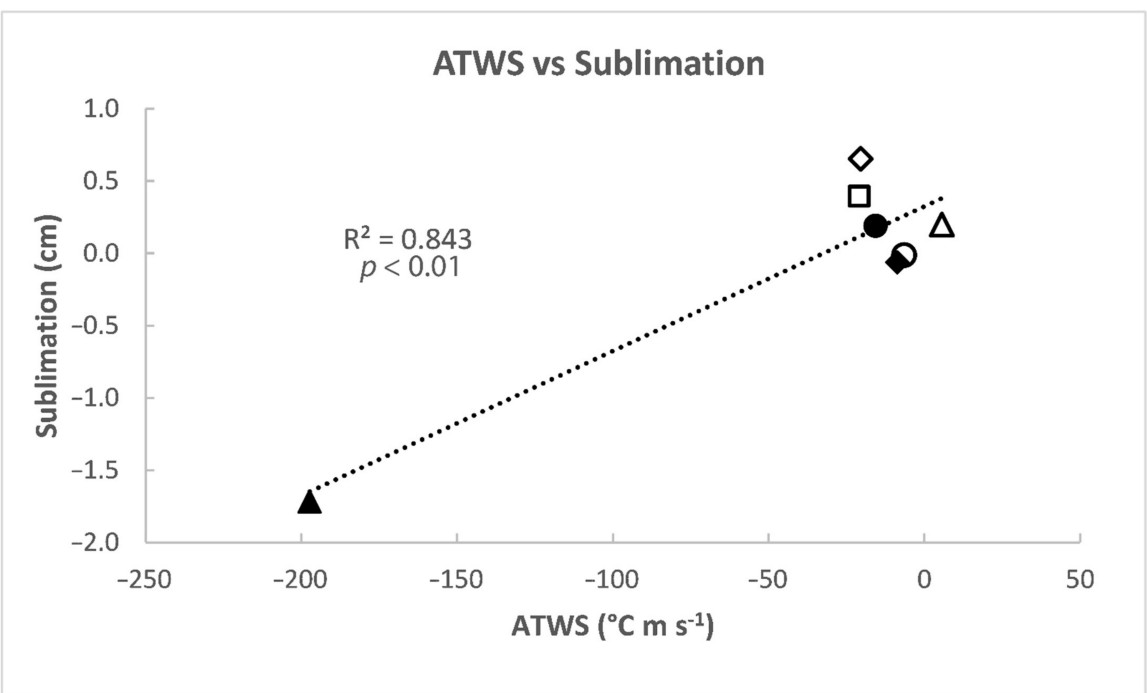

**Figure 7.** Linear regression between the ATWS and the average measured height of all the tubes during the period between 22/11/18 and 16/01/2019 (negative values correspond to loss of ice and positive values correspond to accretion of ice). $R^2 = 0.84$ and $p < 0.01$. The different symbols indicate the date of measurement: black triangle 22/11/18, white square 1/12/18, black rhombus 11/12/18, white triangle 19/12/18, white rhombus 27/12/18, black circle 07/01/19, white circle 16/01/19.

A summary of the methods and the results related to the elevation changes of the lake ice and the frost blisters surfaces of this study is presented in the flowchart (Figure 8). It is noticeable that M1 changes seem almost independent of ATWS and ATDD, suggesting that this frost blister could have a different source of water with respect to the frozen lake and M2.

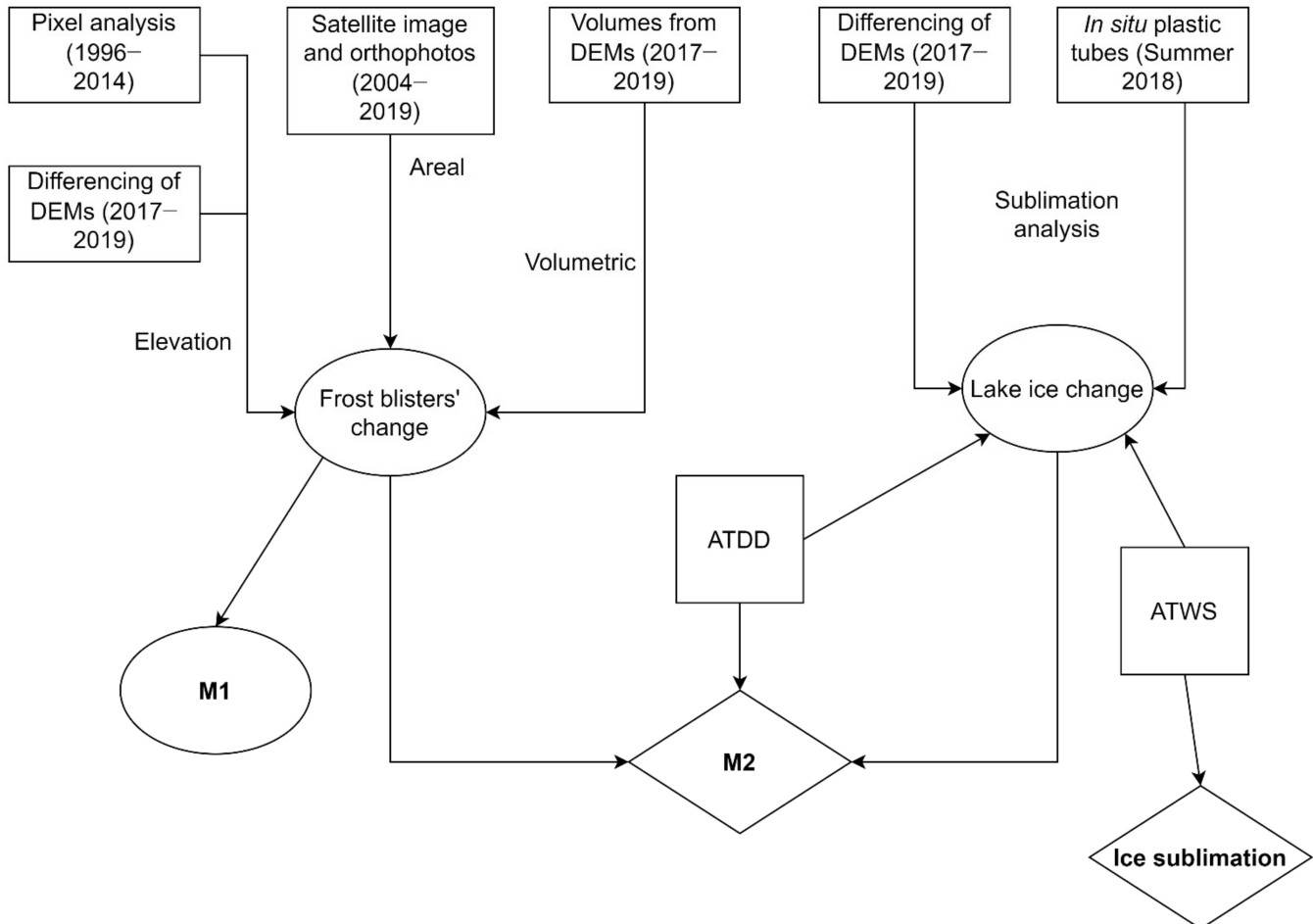

**Figure 8.** Flowchart of the methods and climatic parameters utilized in this paper. Input techniques with the reference period are presented as rectangles, while the derived results are showed as circles. Climatic indices that drive the frost blisters (M2) and ice level change (ATDD, ATWS) are shown as squares. Ice sublimation and ice level are correlated with climatic indices (ATWS for the former and ATDD for the latter). M2 changes are correlated also with both ATDD and ice level while M1 seems to be independent of climatic parameters.

### 4.3. Water Recharge

Concerning M1, it is reasonable to hypothesize that a groundwater flow (of open-system type) supplies this blister. Indeed, the borehole drilled on the side of the blister in 2003 showed the presence of pressurized brines for an amount of 114 kPa, unrelated to the presence of liquids at the lake bottom [10]. Further, a second borehole of M1 in December 2019 (3.94 m of depth, not an object of this study) displayed an average increase of the pressurized brines level in the borehole of 15.9 L h$^{-1}$, greater than the frost blisters that were associated to a hydraulic recharge by van Everdingen [27] in the Arctic.

Moreover in this area, hydrostatic pressure evidences have already been found by French and Guglielmin [11]. One of these consists in algal mats that emerge from the ice surface indicating the presence of refreezing water at the bottom [11]. Recently, the observation of brines beneath the perennially frozen surface at different levels within the lake ice, generally below the icing blisters [63], confirmed the hydrostatic system for this

perennially frozen lake. On the contrary, pressurized brines below M1 suggest a hydraulic injection of water coming from an open-system talik similar to the model described by Forte et al. [20] or Scheidegger and Bense [36]. Independently of the hydrogeologic system, according to previous studies, groundwater recharge is possible through two processes: (a) through talik [26] or sublake open taliks [38] or (b) active layer meltwater infiltration [22] that recharge the supra-permafrost aquifer [38]. In this case, we can surely exclude the second one because we never found water in the shallow boreholes around the lake.

Differently, concerning M2 we cannot exclude the ice uplift through the hydrostatic hypothesis that consists in the surface heaving in early winter as consequence of the meltwater refreezing that percolated through the lateral "moats" (melted edges of lakes in summer, [24]) of the lake as discussed in Guglielmin et al. [11]. In fact, the ice level change may be connected to ice segregation of liquid water at the bottom during autumn [64] likely produced by the solid-state-greenhouse effect [65] or coming from the fusion of snow/ice in the lateral "moats" [10,24]. Moreover, from the most detailed orthophotos (B, C, D), the absence of new cracks on the two icing blisters suggested no hydraulic pressure of the liquid at the bottom in the recent years (2018–2019) confirmed also by a borehole drilled in summer 2019 that found nonpressurized brines on the icing blister close to M1.

Eventually, the future scenario of the lake dynamic will likely follow the trend of an open system because the recent increase of the ground surface temperature [66] will lead an increase of the lake bottom temperature and the formation of through taliks [36]. While the active layer thickening [42] could be related to a drainage of water [16,37] coming from percolating snow melting [22,24], thus leading to a supra-permafrost aquifer seasonal flow [38].

## 5. Conclusions

The photogrammetric surveys performed during the summers 2017–2018 and 2018–2019 allowed us to build the digital elevation models (DEMs) and obtain orthoimages of the perennially frozen lake. This study demonstrated how these remote sensing results are fundamental to determine the spatial and temporal variations of the sublimation and accretion in this peculiar hydric and cryotic system. Here we demonstrated how remote sensing and unmanned aerial vehicles (UAVs) in particular can be a valid tool for assessing the dynamics of cryotic features as frost blisters and to monitor the surface changes and the sublimation rates on perennially frozen lakes.

We demonstrated that an accretion rate is present during summer and it is greater for the frost blisters than the lake ice. Moreover, similar features like two frost blisters in the same lake can have different dynamics and, above all, these differences in surface changes can reflect different water supply mechanisms. Indeed, while M1 seems not to be influenced by climatic indices and revealed the occurrence of pressurized brines within, M2 shows good relations with the ice level and the climatic indices. These evidences suggest the existence of an open-system groundwater flow (hydraulic system) that supply M1, while, for M2, the water supply can be referred to a hydrostatic system as well as for the ice level. Indeed, the hydrostatic hypothesis for the lake ice uplift has been demonstrated [10] and confirmed by the presence of nonpressurized brines underneath the icing blister near to M1. However, the amount of available water would not be able to produce such variations on M2 and on the ice level during the whole examined period, especially with such intense ice sublimation recorded in the last year and therefore a hydraulic open-system could be evoked at least in some years also for M2. Further, the sublimation on the lake ice at Boulder Clay is strongly dependent on the wind speed and low air temperature, and is at its maximum during the Antarctic winter. This is confirmed by the sublimation of the lake ice recorded in situ and by remote sensing that shows an annual ice loss up to 18.1 cm against 0.7 cm during the summer.

**Author Contributions:** Conceptualization, M.G. and S.P. (Stefano Ponti); methodology, S.P. (Stefano Ponti); software, S.P. (Stefano Ponti) and R.S.; validation, S.P. (Stefano Ponti), R.S. and S.P. (Samuele Pierattini); formal analysis, S.P. (Stefano Ponti); investigation, all authors; resources, S.P. (Stefano

Ponti), R.S. and S.P. (Samuele Pierattini); data curation, S.P. (Stefano Ponti), R.S. and S.P. (Samuele Pierattini); writing—original draft preparation, S.P. (Stefano Ponti); writing—review and editing, M.G.; visualization, S.P. (Stefano Ponti); supervision, M.G.; project administration, M.G.; funding acquisition, M.G. All authors have read and agreed to the published version of the manuscript.

**Funding:** This research was funded by PNRA, grant number PNRA16_00194-A1 "Climate Change and Permafrost Ecosystems in Continental Antarctica" and PNRA18_00186-E "Interactions between permafrost and ecosystems in Continental Antarctica".

**Institutional Review Board Statement:** Ethical review and approval were waived for this study, due to studies not involving humans or animals.

**Informed Consent Statement:** Informed consent was obtained from all subjects involved in the study.

**Data Availability Statement:** The data presented in this study are available on request from the first author.

**Acknowledgments:** We want to thank the ENEA logistical support in the frame of PNRA and in particular the Projects "PNRA 2013/AZ1.05 "Permafrost ecology in Victoria Land"; 274 PNRA16_00194-A1 "Climate Change and Permafrost Ecosystems in Continental Antarctica"; 275 PNRA18_00186-E" Interactions between permafrost and ecosystems in Continental Antarctica that allowed this research and Paolo Ciavola for his comments and suggestions.

**Conflicts of Interest:** The authors declare no conflict of interest.

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
