# Peer review of "The Spatio-Temporal Variability of Frost Blisters in a Perennial Frozen Lake along the Antarctic Coast as Indicator of the Groundwater Supply"

_remotesensing, doi:10.3390/rs13030435_

Round 1

Reviewer 1 Report

The article presents the research on frost blister dynamics. The topic is quite interesting but the quality of presentation in low. English language requires extensive editing (improper use of language, grammar mistakes) and in many case the message is unclear. Single-sentence paragraphs should be avoided.

Detailed comments are put on the manuscript. Here are the major ones (copied from the comments pasted on the manuscript):

A good map with clearly shown M1 and M2 is missing. Certain abbreviations are not explained (DJF, JJA).

Why periods between the surveys were not the same? It should be somehow justified in the paper. Why did you need two surveys in Dec. 2018? 

Table 1 is poorly designed.

Raw image processing into DEM should be briefly described.

Why don't we see the DEMs? Why elevation differences were not determined on DEMs?

What is the argument that the changes in relief (visible on DEMs) were due to sublimation? 

Figure 4 caption must be more informative. Which is M1 and M2?

Legend to the map in Fig. 5 is not good.

The arguments supporting the notion of different dynamics of M1 and M2 should be better discussed.

Figure 6 is improper or its caption is improper.

Author Response

Dear Reviewer,

Thank for your efforts in order to improve our manuscript. We did almost all the changes that you requested and addressed your points. Our detailed responses are in italic below while changes in manuscript are marked in red.

The article presents the research on frost blister dynamics. The topic is quite interesting but the quality of presentation in low. English language requires extensive editing (improper use of language, grammar mistakes) and in many case the message is unclear. Single-sentence paragraphs should be avoided.

Detailed comments are put on the manuscript. Here are the major ones (copied from the comments pasted on the manuscript):

A good map with clearly shown M1 and M2 is missing. Certain abbreviations are not explained (DJF, JJA).

You are right, we implemented Figure1 with a helicopter view of the mounds and we changed the Antarctica map with a better one. Abbreviations were explained.

Why periods between the surveys were not the same? It should be somehow justified in the paper. Why did you need two surveys in Dec. 2018? 

We can understand your doubts. Unfortunately, the harsh climate of Antarctica and the logistic difficulties does not permit to follow the schedule of surveys (both helicopter and UAV), therefore we were able, at least to maintain the same survey period that generally represents the warmest period of the year.  In 2018 we did 2 surveys in order to understand also the seasonal variations in the warmest month.

Table 1 is poorly designed.

Thank you, we corrected the spacings and the columns widths.

Raw image processing into DEM should be briefly described.

Thank you. We added at new line 116 a sentence regarding the bundle adjustment that produced the DEMs.

Why don't we see the DEMs? Why elevation differences were not determined on DEMs?

You are right. We did not provide the manuscript with images of DEMs because were too many (4) to be clearly represented in one figure and with such little changes compared to the surrounding topography that they would not be appreciable. However, all the most recent (since 2017, when UAV started to be available) elevation differences were computed on the DEMs.

What is the argument that the changes in relief (visible on DEMs) were due to sublimation? 

We claim that the ice surface decrease is only possible through sublimation, since surficial meltwater was only observed once in January 2002 in the last 20 years except for some slight melting along the shorelines for a few days in a few years.

Figure 4 caption must be more informative. Which is M1 and M2?

You are right. We added the information on the position of M1 and M2 in the caption.

Legend to the map in Fig. 5 is not good.

Thank you, we changed the symbols of the legend in order to make it clearer.

The arguments supporting the notion of different dynamics of M1 and M2 should be better discussed.

Thank you for this suggestion. We implemented the discussion of the blisters’ dynamics in general and also at new lines 293-296.

Figure 6 is improper or its caption is improper.

We agree with you, we modified the caption since it was misleading.

In addition all the other points rised in the annotated manuscript were assessed.

Reviewer 2 Report

The manuscript entitled “The remote sensing techniques as tool for the monitoring of temporal and spatial variability of coastal Antarctic frost blisters and ice surface of a perennial frozen lake as indicators of the 6 groundwater supply” is a very interesting paper of a topic absolutely suitable within the scopes of the journal “Remote Sensing”. The manuscript focuses on one of the perennially frozen lakes and deals with the quantification of the temporal variation of two frost blisters occurring in this lake and the spatio-temporal variations of the ice surface. The paper reads well! It is well written and generally well structured. The overall quality of the paper in terms of methodology and results is high. Here are some comments and suggestions which in my opinion should improve the quality and the structure of the final version of the paper: The title of the paper is too long. The authors should shorten it. The “study area” section should not be part of the “Materials and Methods”. It should be a separate section after the “Introduction”. Regarding the methodology followed a flow diagram of the steps of the approach would be good for the reader. The size of the letters and symbols at the diagram of figure 7 are too large.

Author Response

Dear reviewer,

We want to thank you for your revision that surely improved significantly our papee. Below our reply point to point in italic while changes in the manuscript are marked in red.

The manuscript entitled “The remote sensing techniques as tool for the monitoring of temporal and spatial variability of coastal Antarctic frost blisters and ice surface of a perennial frozen lake as indicators of the 6 groundwater supply” is a very interesting paper of a topic absolutely suitable within the scopes of the journal “Remote Sensing”. The manuscript focuses on one of the perennially frozen lakes and deals with the quantification of the temporal variation of two frost blisters occurring in this lake and the spatio-temporal variations of the ice surface. The paper reads well! It is well written and generally well structured. The overall quality of the paper in terms of methodology and results is high. Here are some comments and suggestions which in my opinion should improve the quality and the structure of the final version of the paper: The title of the paper is too long. The authors should shorten it. The “study area” section should not be part of the “Materials and Methods”. It should be a separate section after the “Introduction”. Regarding the methodology followed a flow diagram of the steps of the approach would be good for the reader. The size of the letters and symbols at the diagram of figure 7 are too large.

Thank you very much for your suggestions. We shortened the title, that now is “The spatio-temporal variability of frost blisters in a perennial frozen lake along the Antarctic coast as indicator of the groundwater supply” and we added a flowchart as suggested and resized the letters of Figure 7. Anyway, we could also change the position of the study area but we read the instructions of the Journal that suggests to locate it in that position and therefore we think that probably is better to leave the chapter in the original position.

Reviewer 3 Report

Dear Authors,

please find my comments on your manuscript.

Author Response

Dear Reviewer thank you for your effeort to improve the paper.

Below you can find the replies point to point to your report where our replies are in italic whereas in the manuscript the changes are marked in red.

General comments:

Thank you for your suggestions. Paragraphs, english phrasings and references to the figures have been corrected or implemented.

Abstract and Keywords:

The corrections have been made.

Introduction:

We agree with you. We made corrections to the misleading parts and we implemented the key point of the introduction with a brief description of the studies cited.

Materials and methods:

We adjusted the errors you mentioned. More importantly, we moved part of the discussion here to explain in the proper section the climatic analyses and we merged the subchapter 2.1 with 2.2 as requested.

Results:

You are right. We implemented the climatic results with the descriptions of the trends through the equations.

Discussion:

We provided Figure 7 with the R-square and p values and moved the description of Figure 6 in the text (discussion). Moreover, we moved the sentences concerning the methods of analysis in the Materials and Methods section.

Conclusion:

Thank you for your comment. We extended the conclusions according to our results.

Round 2

Reviewer 1 Report

Dear Authors,

thank you for the correcions, the article is much better know and I recommend it for publication.

Regards

MD

Author Response

Dear Reviewer thank you so much for your further check. Your efforts were really appreciated.

Mauro

Reviewer 3 Report

Dear Authors,

after reading the changes that you have incorporated into the manuscript I believe that it is now ready for being published.

However, before that happens you must edit your manuscript according to the journal format. Also, read carefuly the manuscript to avoid places like lines 377-382 which are not fulfilled.

Other than that, well done.

Kind regards

Reviewer

Author Response

Dear Reviewer, thank you for your further effort in checking the manuscript.

Our reply is here in italic while the changes in the manuscript are marked in red.

after reading the changes that you have incorporated into the manuscript I believe that it is now ready for being published.

THank you so much

However, before that happens you must edit your manuscript according to the journal format. Also, read carefuly the manuscript to avoid places like lines 377-382 which are not fulfilled.

We reformatted the paper according the new format that we found as template and we hope to have understood correctly what you requested for lines 377-382.